# Virtual Reality for PEripheral Regional Anesthesia (VR-PERLA Study)

**DOI:** 10.3390/jcm9010215

**Published:** 2020-01-13

**Authors:** Camille Alaterre, Baptiste Duceau, Eileen Sung Tsai, Siham Zriouel, Francis Bonnet, Thomas Lescot, Franck Verdonk

**Affiliations:** 1Department of Anaesthesiology and Intensive Care, Hôpital Saint-Antoine, Assistance Publique-Hôpitaux de Paris, 75012 Paris, France; 2Department of Anaesthesiology and Intensive Care, Hôpital européen Georges-Pompidou, Assistance Publique-Hôpitaux de Paris, 75015 Paris, France; 3Department of Anesthesiology, Perioperative and Pain Medicine, Stanford University School of Medicine, Stanford, CA 94305, USA; 4Sorbonne University, 75006 Paris, France

**Keywords:** anesthesia, regional anesthesia, virtual reality, ambulatory surgery, satisfaction, anxiety

## Abstract

When used as an add-on to regional anesthesia, virtual reality (VR) has been reported to provide anxiety-reducing benefits and sedation-sparing effects. However, its impact on patient satisfaction is still a matter of controversy. We investigated the feasibility and benefits of implementing intraoperative VR distraction in a French University Hospital (Hôpital Saint-Antoine, AP-HP). This monocentric observational before–after study included 100 patients who underwent ambulatory upper limb surgery under peripheral nerve block in January 2019, 50 before and 50 after implementation of an intraoperative VR distraction protocol. Primary outcome was patient self-rated satisfaction score evaluated right after surgery. Secondary outcomes included 2-month patient-reported satisfaction score, perioperative self-rated anxiety and intraoperative hemodynamic changes. Compared to former standard care, VR distraction was associated with significantly higher postoperative satisfaction scores (10 [IQR 9; 10] vs. 9 [8; 10], *p* < 0.001) still reported two months after surgery (10 [10;10] vs. 10 [8.5;10], *p* = 0.06). Patient median intraoperative anxiety score was lower in the VR group, compared to Standard Care group (0 [0; 2] vs. 3 [0.25; 7], *p* < 0.001), and occurrence of intraoperative hemodynamic changes was also lessened in the VR group (2% vs. 16%, 0R = 0.11[95% CI 0.002–0.87], *p* = 0.031). The present findings suggest that VR distraction program in the operating room could effectively improve patient satisfaction with anxiety-reduction and hemodynamic benefits.

## 1. Introduction

Regional anesthesia (RA) allows surgery without impacting the patient’s level of consciousness. This avoids potential risks and disadvantages related to general anesthesia, especially in regards to control of upper airways. It is, therefore, not surprising that RA techniques and especially peripheral nerve blocks have turned out to be essential for the development of ambulatory surgery [1]. However, the prospect of “hearing and seeing everything” in the operating room can still be a source of great anxiety and discomfort for the patient, thus modulating pain perception [2], leading to dissatisfaction and possibly failure of the regional anesthesia technique [3].

Several distraction tools have been shown to be useful in reducing anxiety, decreasing pain levels, and improving patient satisfaction outside the operating room [4,5]. Virtual reality (VR) appears to be one of these effective diversion methods, having proved its benefits especially in the care of burned patients [6,7], patients with breast cancer [8], and pediatrics patients [9,10]. By simulating the physical presence of the user in an artificially generated environment, VR can modulate the activation of several brain areas, especially the anterior cingulate cortex, insula, and tonsil, which are involved in attentional and emotional pain pathways [11,12].

Once expensive and cumbersome, VR is now accessible to a wider audience by means of smaller, more comfortable, and easy-to-use devices, usable in the perioperative care. VR was first investigated by perioperative practitioners as a simulation tool, for pedagogical and relaxing pre-operative immersion sessions aiming at reducing patient anxiety before undergoing general anesthesia [13,14] and while performing peripheral nerve block to lower noxious stimulations [15]. More recently, two studies have suggested that it would be possible to use a distractive virtual reality headset inside the operating room, during surgical procedures performed under regional anesthesia, with benefits in terms of anxiety [16] and sedation sparing effects [17]. A recent study published in 2019 investigating 37 patients undergoing urological endoscopic procedures under spinal anesthesia reported that using virtual reality during surgery could improve patient and anesthesiologist postoperative satisfaction scores compared to usual sedation with midazolam [18]; these results have not yet been confirmed in orthopedic surgery [17], and peripheral regional anesthesia remains under-studied. Moreover, there is no information available on potential long-term satisfaction and hemodynamic benefits of VR distraction technique. Surgical procedures practiced under peripheral nerve blocks are yet becoming more and more common, especially with the development of the outpatient management [19], and patient satisfaction at short and long term appears more than ever as a central marker of the quality of health care [20].

The current study has investigated the feasibility and benefits in terms of patient satisfaction when implementing intraoperative VR distraction during surgical procedures performed under peripheral regional anesthesia. The secondary objectives were to assess the impact of VR on patient anxiety, hemodynamic changes associated with surgery, RA failure rate, analgesic consumption, and length of stay in the post-anesthesia care unit (PACU).

## 2. Methods

### 2.1. Study Design

Since 14 January 2019, intraoperative VR distraction has been offered to patients operated in the Ambulatory Surgical Unit of our institution as a new standard care protocol. In order to evaluate this practice, this monocentric observational before–after study compared medical data, self-rated satisfaction and anxiety scores of 100 patients who underwent ambulatory upper limb surgery under peripheral nerve block in January 2019, over the two periods defined as “before” and “after” the VR protocol implementation. This determined respectively, the “Standard Care” study group and the “Virtual Reality” study group.

### 2.2. Patients

Eligible patients were all patients aged 18 years or older who had scheduled or emergency upper limb ambulatory orthopedic surgery under peripheral nerve block, between 1 January 2019 and 31 January 2019. The non-inclusion criteria were: Cognitive impairment; non-French speaking or understanding; a history of epilepsy, psychosis, or claustrophobia; blindness or deafness; and refusal to use the headset if applicable.

### 2.3. Study Protocol

In order to assess the implementation of intraoperative VR, a satisfaction survey was conducted with 100 similar post-operative questionnaires issued. Fifty patients who did not have access to VR were included from 1 to 13 January (Standard care group) and fifty patients who benefited from VR distraction during their surgery were included from 14 to 31 January (VR group). Questionnaires consisted of three visual 10-point graduated scales [21] assessing immediate postoperative satisfaction, anxiety level before surgery, and anxiety level during surgery (See Appendix B
Figure A2). In any case, they were given to patients after surgical procedure on arrival in the Post-Anesthesia Care-Unit (PACU) and retrieved before discharge.

Included patients, as all patients taken care of in our center, had benefited from a peripheral nerve block performed by the anesthetist physician in the pre-operative holding area. The type of nerve block depends on the intervention foreseen, and the choice is usually left to the discretion of the senior anesthetist in charge. Use of sedation for reducing pain and anxiety during nerve block technique is also left to the discretion of the anesthetist physician. All nerve blocks are tested for efficacity before patient’s entrance in the operating room but additional intraoperative sedation or even conversion to general anesthesia is possible on a case-by-case basis, if deemed necessary.

Medical records of all included patients were collected retrospectively to obtain demographic data, ASA physical status scores, basic parameters, anesthetic technique, type and duration of surgery, intraoperative monitored parameters, intraoperative sedation requirement, analgesic demand, and postoperative data such as PACU length-of-stay and post-operative pain scores evaluated by the Numerical Rating Scale (NRS). In the VR group, time required to set-up the virtual reality device, session lengths, early discontinuation, adverse events, and practitioners (anesthesiologists and surgeons) satisfaction scores were also collected from specific monitoring sheets (See Appendix B
Figure A3 and Figure A4). All included patients were contacted by a phone call in March 2019, i.e., two months after surgery, to be questioned again about their overall satisfaction with regional anesthesia. Those data were integrated into a secured database and analyzed.

### 2.4. Intraoperative Virtual Reality

The device was a stand-alone “Oculus Go 64 Giga Octets” virtual reality headset from OCULUS VR^®^ (San Francisco, CA, USA) measuring 190 × 105 × 115 mm and weighing 467 g with a resolution of 2560 × 1440 pixels, connected to a “H840” model audio headset from EDIFIER^®^ (Beijing, China), both used with protections validated by the hospital’s hygiene team (Figure 1). From 14 January 2019, in absence of existing contraindication, virtual reality was presented to the patient by the anesthetist team (doctor or nurse) before anesthesia and surgery in the pre-operative holding area. The existence of contraindications or refusal were recorded if applicable. Whenever accepted, the helmet was set up by the anesthesiologist or anesthetist nurse after installation in the operating room and removed at the end of the surgical procedure. The session could be interrupted prematurely at any time for medical reasons or on patient’s request. Early discontinuations and adverse events were recorded in the specific monitoring sheet (See Appendix B
Figure A3 and Figure A4). The first-line content offered was an immersion into a relaxing natural environment downloaded from the free “Guided Meditation VR” application, developed by CUBICLE NINJAS AGENCY^®^ (Glen Ellyn, IL, USA) and available for free access on the download platform (oculus store). Patients could choose their environment from the following selection: Relaxing day on a tropical beach, beautiful sunset beach, mountain sunrise, or forest nap experiences (See Appendix A
Figure A1). A distracting video entitled “Documentary on the Return of the Lion,” previously downloaded from a PC was offered as a second intention.

### 2.5. Study Outcomes

The primary outcome was postoperative patient-reported satisfaction score, assessed on a visual 10-point satisfaction scale [22] listed in the questionnaire filled in PACU (see Appendix B
Figure A2). Patients were defined as “very satisfied” if their rating was greater than or equal to nine out of ten.

Secondary outcomes included 2-month patient-reported satisfaction, perioperative anxiety scores, occurrence of regional anesthesia failure, intra and postoperative analgesic consumption, and pain scores on PACU arrival. Two-month 10-point satisfaction scores were verbally reported at the end of a telephone interview two months after surgery. Anxiety scores were assessed by visual pre-operative and intraoperative anxiety scales graduated from zero to 10 [16] filled-in retrospectively in the PACU questionnaire (see Appendix B
Figure A2). RA failure was defined as the use of a complementary intraoperative sedation. Pain scores were assessed by the Numerical Rating Scale (NRS).

Hemodynamic outcomes included occurrence of intraoperative hemodynamic changes [23], tachycardia, bradycardia, hypertension, hypotension, or decrease in pulse oxygen saturation episodes. Hemodynamic changes were defined as an increase of 30% in systolic blood pressure and/or heart rate during surgery compared to baseline. Tachycardia was defined as a heart rate (HR) greater than 90 beats per minute (bpm). Bradycardia was defined as HR less than 50 bpm. Hypertension was defined as systolic blood pressure (SBP) greater than 150 mmHg. Hypotension was defined as blood pressure defined as SBP less than 90 mmHg. Desaturation was defined as pulse oxygen saturation (SpO2) less than 90%.

Feasibility and security outcomes in the VR study group included: median time required to set-up the device; VR session lengths; occurrences of early stops; occurrences of device-related complications such as headache, nausea, or vomiting; proportion of patients wishing to use the technology again for future surgeries; and practitioners (anesthesia and surgery teams) satisfaction scores assessed by visual 10-point scales.

### 2.6. Regulatory and Ethical Issues

In accordance with the French “Jardé Law” on biomedical research, this observational before–after study, combining a satisfaction survey with a retrospective analysis of data collected as a part of usual care, fell into the categories of “research studies not involving human subjects” [24] and thus obtained the approval of the French Institutional Review Board “Comité d’Éthique de la Recherche en Anesthésie-Réanimation” (CERAR, president Pr JE Bazin, 12 May 2019) under the reference IRB-00010254-2019-071. Patients were all institutionally informed via the websites of the Assistance Publique-Hôpitaux de Paris (AP-HP) and Saint-Antoine Hospital of the possible use of their data in researches aimed at improving the quality of care, as well as their right and terms of objection. This information was also included for each patient in the hospital’s welcome booklet, given on administrative registration, and presented at the end of the hospitalization reports.

In order to guarantee the security of personal data, the investigators retrospectively collected and integrated the information anonymously into a secure database, in accordance with the French CNIL MR-004 methodology, and registered in the AP-HP processing register under number 20190625112233.

### 2.7. Statistical Analysis

The number of subjects required was calculated considering historical data collected in our center assessing the satisfaction of patients who had undergone ambulatory surgery under peripheral regional anesthesia in an outpatient circuit in 2018. Those historical data showed that the median satisfaction scores were globally high and did not follow a normal distribution. Preliminary satisfaction data from four patients managed with the VR device during the optimization phase of the VR service protocol showed that satisfaction score seemed higher than without VR device. We used a bootstrapping technique (1000 replicates) to assess the number of patients needed to demonstrate a significant difference between satisfaction scores of patients managed with and without VR. Using a non-parametric test (the Wilcoxon–Mann–Whitney test), with an alpha risk at 0.05 and a power of 90%, we calculated that the inclusion of 89 patients was necessary to show a statistically significant difference on the primary endpoint. To account for 10% of missing data, the inclusion of 50 patients was planned in each group. All analyses were performed on the full analysis set according to the intention-to-treat principle.

Normality of the quantitative variables was assessed by the Shapiro–Wilk test. The quantitative variables are presented as medians (25th; 75th percentiles) or means (standard deviation) according to their distribution. Qualitative variables are presented in terms of occurrence (%). Univariate comparisons between the two groups were made using the exact Fischer test for qualitative variables, the Student’s t-test for quantitative variables with normal distribution, and the Mann–Whitney test for quantitative variables with non-normal distribution. The R Software (version 3.5.1 for Macintosh, GNU GPL licenses, “The R Foundation for Statistical Computing”, Vienna, Austria) was used to perform all statistical analyses. The statistical tests were bilateral and a *p*-value of less than 0.05 was set to define statistical significance.

## 3. Results

### 3.1. Study Groups

One hundred and sixty-four (164) patients aged 18 or older received upper limb orthopedic surgery under peripheral regional anesthesia in the center in January 2019, 74 before and 90 after intraoperative VR implementation. To meet our inclusion objectives, 50 postoperative satisfaction questionnaires were consecutively distributed to eligible patients and then collected during each of the two periods. Among the 74 patients operated on in the first period, the 50 first received and completed the questionnaire. They were included in the analysis as the Standard Care group. During the second period, the first 58 patients were evaluated. Eight patients were excluded: Four patients declined the VR helmet, and four patients had contraindications to its use (three patients had language barriers and one had a history of epilepsy). The first 50 patients who effectively received VR were given the satisfaction questionnaire and were included in the analysis (VR group). No patient, over either study period, refused to complete the questionnaire given to them.

In fine, data from 100 patients were analyzed, 50 in the Standard Care group and 50 in the VR group. Patients were comparable in terms of demographics, ASA scores, types and durations of surgeries, and modalities (type and dose) for regional anesthesia. In most cases, they had benefited from an axillary block, performed fully awaken with no additional midazolam sedation (Table 1). Patients in the VR study group had received intraoperative virtual reality distraction for a median duration of 25 (IQR [19; 37]) minutes.

### 3.2. Primary Outcome

Immediate postoperative satisfaction score was significantly higher in the VR group compared to the Standard Care group (median satisfaction score = 10 [Interquartile 9; 10] vs. 9 [8; 10], *p* < 0.001, Figure 2) with a significant increase in the proportion of very satisfied patients (*n* = 48 vs. *n* = 32, (*p* < 0.001), Odds Ratio = 13.2 [95% CI 2.8–125.1]).

Eighty-four percent (84%) of patients could be contacted by telephone two months after the surgical procedure: 41 in the Standard Care group and 43 in the VR group. The benefit persisted over time, with significantly higher satisfaction scores at two months in patients who had virtual reality (median satisfaction score = 10[Interquartile 10; 10] vs. 10[8.5; 10], *p* = 0.006, Figure 2). There was no significant change in the reported satisfaction scores between D0 and M2 within each group.

### 3.3. Anxiety

Pre-operative anxiety scores did not differ between the two groups. However, anxiety level was significantly lower during the procedure in the VR group compared to the Standard Care group (median subjective anxiety score = 0 [0; 2] vs. 3 [0.25; 7], *p* < 0.001). There was a significant reduction in the level of anxiety reported before and during the surgery only in the VR group (median variation = −3.0 [−5.0; −1.0], (*p* < 0.001) Figure 3).

### 3.4. Hemodynamic Outcomes

Baseline vital parameters (SBP, DBP, HR) measured on admission in the ambulatory surgery department did not differ between the two groups. The use of virtual reality was associated with a significant reduction in the occurrence of intraoperative hemodynamic changes (2% vs. 16%, 0R = 0.11[95% CI 0.002–0.87], *p* = 0.031). Tachycardia was also less frequent in the VR group compared to the Standard Care group (10% vs. 28%, 0R = 0.30[95% CI 0.07–0.95], *p* = 0.041, Table 2). There has been no episode of bradycardia, hypotension, or desaturation in either group. There was no correlation between intraoperative anxiety scores and heart rate or blood pressure values.

### 3.5. RA Failure, Analgesic Consumption, PACU Lengh of Stay

The use of complementary intraoperative sedation, defining failure of regional anesthesia, was not significantly different between patients operated with or without virtual reality (4% VR group vs. 8% Standard Care group, *p* = 0.64). There was no difference in perioperative analgesic consumption and no difference in PACU length of stay between the two groups (25 [20; 30] minutes VR group vs. 29 [20; 31] minutes Standard Care group, *p* = 0.50).

### 3.6. Feasability and Tolerance

We studied intraoperative virtual reality as an adjunctive to the peripheral nerve blocks of 50 patients. The median set-up time was 2 (IQR [2; 5]) minutes. For median session duration of 25 (IQR [19; 37]) minutes, no adverse events related to the device such as headache, nausea, or vomiting were reported. One patient wearing glasses was uncomfortable with the helmet of the device and asked to stop the session before the end of the surgical procedure; no medical reason for termination was reported. Ninety-four percent (94%) of patients having benefited from virtual reality indicated that they would like to use this technology again for future surgery under RA. Regarding those interventions performed with virtual reality, median satisfaction score of the anesthesia team was 9 [9; 10] out of 10 meanwhile satisfaction of surgical team reached 10 [9; 10] out of 10.

## 4. Discussion

We have compared two groups of patients who underwent upper limb orthopedic surgeries under peripheral nerve blocks, before and after implementation of an intraoperative virtual reality distraction protocol. Satisfaction scores of patients who received VR as a complement to their regional anesthesia were significantly higher than those of patients operated under regional anesthesia alone, in both the short and long term. There was a significant increase in the proportion of very satisfied patients in the VR study group, with a persistent benefit two months after anesthetic management. Subjective intraoperative anxiety was significantly reduced when virtual reality was used, and anxiolytic effects of the technique were suggested by a significant decrease in patient-reported anxiety scores during surgery compared to the preoperative period when accessing it. Virtual reality was also associated with fewer intraoperative hemodynamic changes in blood pressure and heart rate with reduced occurrence of tachycardia episodes. No adverse event related to the device use was reported during this study.

Nowadays, gathering data on the patient’s satisfaction is critical to assess and improve the quality of health care [20]; to our knowledge, this study is the first one to show the benefits of virtual reality as an add-on to peripheral RA in terms of short and long term patient-reported satisfaction, and in the field of ambulatory orthopedic surgery. Previous studies investigating virtual reality in the operating room have most often been carried out during lower-limb surgeries and compared VR to sedation protocols. Few of them investigated VR for procedures performed under peripheral regional anesthesia alone, though these techniques are increasingly being recommended in order to limit the impaired area and promote functional recovery and rapid return of the patient at home [19]. Our results suggest that distractive virtual reality could become a new standard practice and be widely applied to patients undergoing ambulatory orthopedic surgery, as an add-on to peripheral regional anesthesia, with benefits in terms of patient satisfaction and reduction in perioperative anxiety levels leading to greater hemodynamic stability during surgery. VR is a simple and affordable distraction technique that can be used by everyone and everywhere; its implementation can be readily effective in many surgery centers around the world. Although imperfect comfort has been reported when using this model of helmet in supine position, especially for patients wearing eyeglasses, our study did not report any complications such as headache, nausea, or vomiting, nor did we see any premature medical arrest. Most of the patients (94%) would like to repeat the virtual reality experience in the event of new surgery. Practitioners were also very satisfied with the arrival of this technology, with setting-up times compatible with the ambulatory surgery block activity.

The feasibility and the safety of VR are supported by the study of Chan et al., who demonstrated in 9 patients undergoing hip or knee replacement surgery under regional anesthesia that it was possible to use a virtual reality helmet during the procedure [17]. They have shown a non-significant sedation-sparing effect in the VR group but no benefit in terms of postoperative patient satisfaction. However, in this small study, each of the two groups systematically received a sedation protocol, explaining the lack of difference observed. A recent study by Jee Youn Moon et al. supports our data in terms patient satisfaction; they showed a higher proportion of very satisfied patients when using a virtual reality helmet for urological surgical procedures performed under spinal anesthesia, compared to a sedation protocol [18].

By evaluating 50 patients who received intraoperative VR immersion, we obtained feasibility and safety data much more strongly than previous studies, that included a maximum of 20 VR patients each. Outpatient surgery is a sector that has experienced strong growth in recent years, with significant investments by hospital groups and efficiency needs that require practices optimization. In outpatient surgery performed under regional anesthesia, the management of a patient’s anxiety is of utmost importance. Indeed, the need for intraoperative complementary sedations is considered a partial failure of the technique and can delay discharge from hospital, a major quality standard [1]. Although not statistically significant, the use of complementary intraoperative sedation in our study was reduced from 8% to 4% in the VR group. We infer that VR distraction would tend to prevent the use of complementary sedations during surgery. This appears to us as one potential factor explaining the satisfaction benefits of VR protocol. There was no difference in terms of PACU length of stay, which was consistent with the overall low occurrence of complementary sedations in this peripheral regional anesthesia expert center.

This study has several limitations. First of all, it is an observational and monocentric study. Indeed, we sought to evaluate the impact of a VR protocol and its implementation by a multidisciplinary team. Although randomized controlled trials are the gold standard, conducting an interventional study could have biased our protocol from its clinical reality. The before–after study falls within the category of quasi experimental studies; its limitations notwithstanding, it is useful in situations such as ours in which an interventional study design is hardly conceivable. Of course, it shall not be forgotten that comparative before–after methodology and absence of randomization can generate both selection bias and confounding factors, leading to caution in interpreting the results. However, we took care to control several potential biases; the study was conducted on a short-predetermined period of one month during which there was no change in the practitioners and practices, limiting the risk of history bias. The two groups were comparable for demographics and surgical procedures performed. We kept the “before” and “after” measurement methods constant, giving questionnaires in the same format and under the same conditions for all patient over the two study periods, thus limiting the risk of reporting bias. In both groups, the 50 first patients were given and completed the satisfaction questionnaire; this potential selection bias therefore does not appear as a differential bias. If it did, it would therefore be in favor of the control group and would lead to underestimation of the size of the VR effect [25]. On another matter, it is true that satisfaction scores were on average very high in our study even before VR implementation; it cannot be excluded that increase in patient satisfaction observed when benefiting from intraoperative VR may be the consequence of spending more time with the patients when explaining the modalities and functioning of the device, more than the VR session itself. Furthermore, we chose to assess patient satisfaction by using a visual scale graduated from zero to 10, which is a simple and rapid tool widely used in this type of study [21,22] and not a more objective tool such as the “EVAN-ALR” scale [26]; however, the implementation and the time required to complete the latter were judged not to be compatible with the clinical reality of our ambulatory surgery department. Eventually, we studied orthopedic surgeries of the upper limb exclusively and performed in an ambulatory circuit; these data cannot therefore be extended to other types of procedures whose installations may be incompatible with the use of a virtual reality helmet.

## 5. Conclusions

Implementation of a virtual reality distraction protocol in the operating room as an add-on to peripheral regional anesthesia could effectively improve patient satisfaction and reduce perioperative anxiety, with hemodynamic stabilization effects, without any complication related to the device and with excellent acceptability from the medicosurgical team.

## Figures and Tables

**Figure 1 jcm-09-00215-f001:**
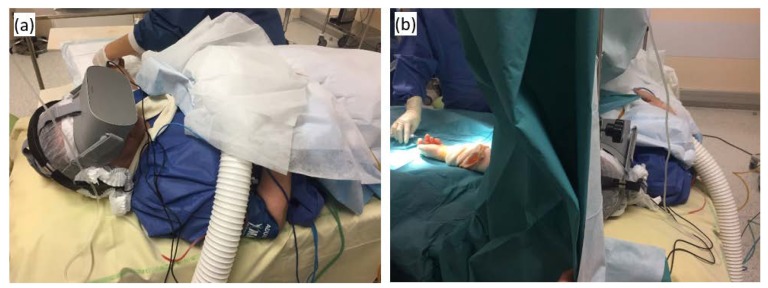
(**a**) Virtual reality helmet set up after installation in the operating room; (**b**) intraoperative virtual reality session.

**Figure 2 jcm-09-00215-f002:**
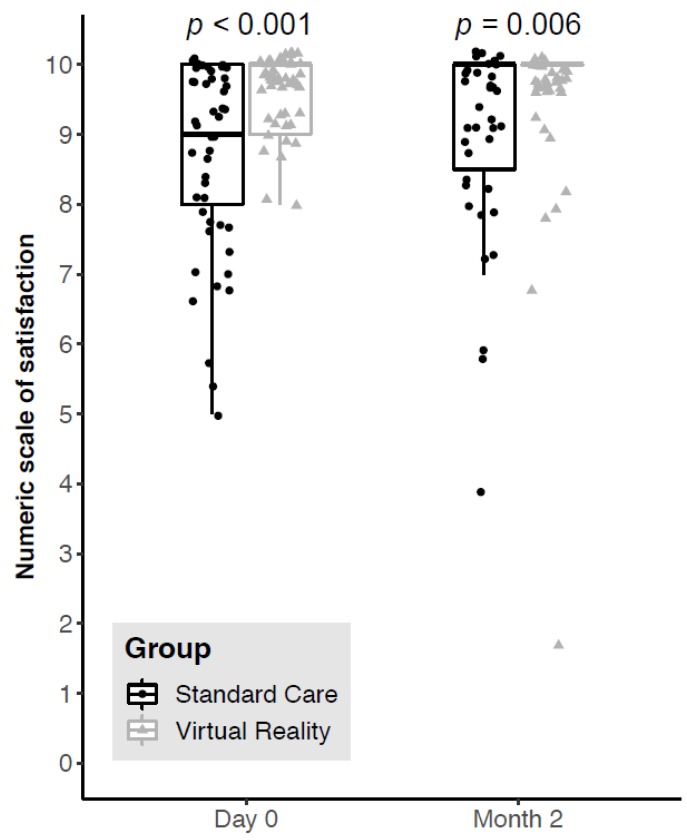
Boxplot graphical representations of the immediate postoperative and 2-month postoperative 10-points satisfaction scores of patients operated before (Standard Care group, dark-grey dots boxplot) and after (Virtual Reality group, light-grey triangles boxplot) virtual reality became available in the department. The upper edge of the box represents the 75th percentile and the lower edge represents the 25th percentile. The vertical length of the box represents the interquartile interval and the central horizontal line represents the median. The upper moustache extends from the upper edge to the highest value at 1.5 times the interquartile space. The lower moustache extends from the lower edge to the lowest value at 1.5 times the interquartile space each dot or triangle represents a patient. Some jittering was added to prevent the overplotting of dots. Please note that in the VR study-group, at month 2, the representation of the boxplot merges with its median.

**Figure 3 jcm-09-00215-f003:**
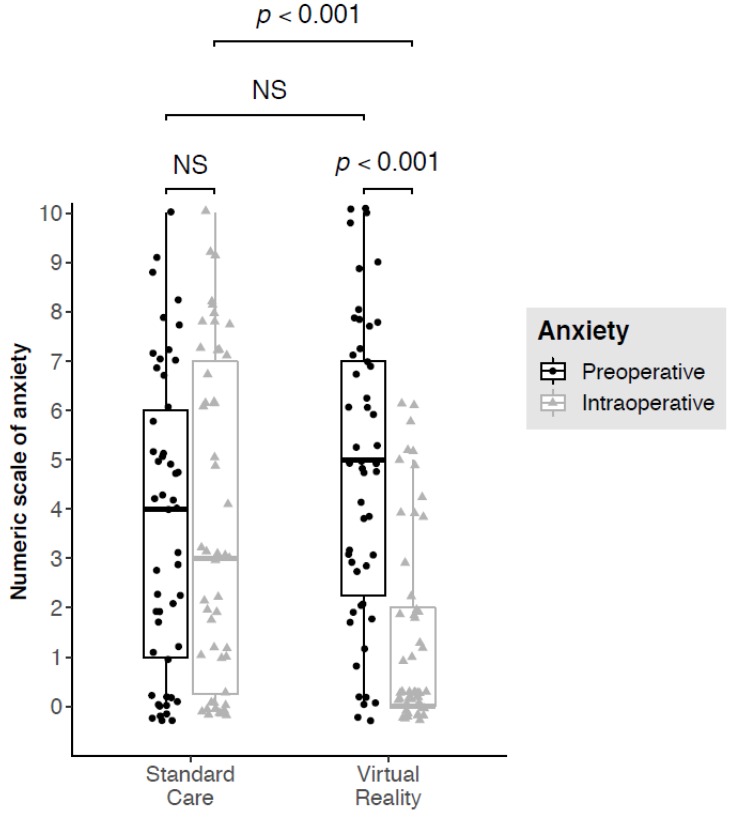
Boxplot graphical representation of perioperative (pre—in dark-grey plots and intraoperative—in light-grey triangles) anxiety scores of patients operated before (Standard Care group) and after (Virtual Reality group) virtual reality became available in the department. The upper edge of the box represents the 75th percentile and the lower edge represents the 25th percentile. The vertical length of the box represents the interquartile interval and the central horizontal line represents the median. The upper moustache extends from the upper edge to the highest value at 1.5 times the interquartile space. The lower moustache extends from the lower edge to the lowest value at 1.5 times the interquartile space. Each dot or triangle represents a patient. Some jittering was added to prevent the overplotting of dots.

**Table 1 jcm-09-00215-t001:** Demographic, surgical and anesthetic data of patients operated before (Standard Care group) and after (virtual reality (VR) group) the virtual reality headset became available.

Variables		Standard Care Group (*n* = 50)	VR Group (*n* = 50)	*p*-Value
Sex, *n* (%)	Women	18 (36)	24 (48)	0.311
	Men	32 (64)	26 (52)	
Age, mean (SD)		49 (19)	48(19)	0.895
ASA score, *n* (%)	1	38 (76)	30 (60)	0.215
	2	11 (22)	19 (38)	
	3	1 (2)	1 (2)	
Surgical procedure, *n* (%)	Wound	10 (20)	11 (22)	0.911
	Carpal tunnel	9 (18)	7 (14)	
	Infection	9 (18)	8 (16)	
	Material removal	7 (14)	4 (8)	
	Trigger finger	4 (8)	6 (12)	
	Fracture	4 (8)	5 (10)	
	Dupuytren	3 (6)	2 (4)	
	Other	4 (8)	7 (14)	
Surgical site, *n* (%)	Forearm	3 (6)	1 (2)	0.726
	Arm	1 (2)	1 (2)	
	Wrist	19 (38)	18 (36)	
	Hand	6 (12)	10 (20)	
	Finger	21 (42)	20 (40)	
Planification, *n* (%)	Scheduled	28 (56)	24 (48)	0.841
	Emergency	22 (44)	24 (48)	
Duration of surgical procedure (min), median [IQR]		30 [23; 43]	32 [25; 40]	0.822
RA technique, *n* (%)	Distal Block(s)	2 (4)	0 (0)	0.475
	Axillary block	48 (96)	50 (100)	
Local anesthetic agent, *n* (%)	Xylocaine 15 mg/mL	49 (98)	48 (96)	1.000
	Ropivacaine 3.5 mg/mL	1 (2)	2 (4)	
Local anesthetic doses (mg), median [IQR]	Xylocaine	315 [300; 360]	300 [300; 375]	0.221
	Ropivacaine	35 [35; 35]	79 [77; 79]	
Planned adjunctive analgesic distal nerve block, *n* (%)	Median nerve block	2 (4)	1 (2)	0.053
	Ulnar nerve block	3 (6)	0 (0)	
	Radial nerve block	2 (4)	0 (0)	
	Radial + Median nerve blocks	3 (6)	8 (16)	
	Radial + Ulnar nerve blocks	2 (4)	0 (0)	
Per-RA technique Midazolam sedation, *n* (%)	YES	2 (4)	1 (2)	1.000
	NO	48 (96)	49 (98)	

Values are expressed as mean (standard deviation), median (interquartile deviation) or actual (%). The *p*-value results from an exact Fischer test for qualitative variables and a Mann–Whitney test for quantitative variables whose distributions were not normal. Abbreviations: LA = local anesthetics; RA = regional anesthesia; ASA = American Society of Anesthesiologists physical status classification; SD = standard deviation; IQR = interquartile range; mg = milligrams; min = minutes.

**Table 2 jcm-09-00215-t002:** Perioperative hemodynamic vital parameters.

Variables		Standard Care Group (*n* = 50)	VR Group (*n* = 50)	*p*-Value
Systolic blood pressure (mmHg), median [IQR]	SBP baseline	135 [125; 140]	140 [125; 150]	0.188
	SBP intraoperative max	140 [130; 158]	135 [127;143]	0.106
	SBP intraoperative min	122 [118; 132]	120 [114; 132]	0.512
Heart rate (bpm), median [IQR]	HR baseline	75 [65; 85]	70 [65; 80]	0.459
	HR intraoperative max	85 [80; 91]	75 [70; 82]	<0.001
	HR intraoperative min	74 [68; 81]	69 [60; 75]	0.069
Intraoperative hypertension, *n* (%)		17 (34)	9 (18)	0.111
Intraoperative hypotension, *n* (%)		0 (0)	0 (0)	NA
Intraoperative tachycardia, *n* (%)		14 (28)	5 (10)	0.041
Intraoperative bradycardia, *n* (%)		0 (0)	0 (0)	NA
Intraoperative desaturation, *n* (%)		0 (0)	0 (0)	NA

The values are expressed as median [interquartile range] or actual (%). The *p*-value results from an exact Fischer test for qualitative variables and a Mann–Whitney test for quantitative variables whose distributions were not normal. Hypertension was defined as SBP > 150 mmHg, hypotension as SBP < 90 mmHg, tachycardia as HR > 90 bpm, and bradycardia as HR < 50 bpm. SBP and baseline HR were the values measured at admission in the outpatient surgery department. Abbreviations: bpm = beats per minute; IQR = interquartile range; HR = heart rate; max = maximum; min = minimum; mmHg = millimeters of mercury; NA = not applicable; DBP = diastolic blood pressure; SBP = systolic blood pressure.

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
