# Peer review of "Virtual Reality for PEripheral Regional Anesthesia (VR-PERLA Study)"

_jcm, 2020, doi:10.3390/jcm9010215_

Round 1

Reviewer 1 Report

Thank you for replying to all of the comments and questions posed.  There are a few minor grammatical issues which I think may be best left to the editors to handle, but overall I really found this article to be well done, and the responses to the questions to have clarified any uncertainties in the paper.  
Best of luck!

Reviewer 2 Report

Thank you for your extensive and satisfying answers to my ethical concerns

This manuscript is a resubmission of an earlier submission. The following is a list of the peer review reports and author responses from that submission.

Round 1

Reviewer 1 Report

This is the study about the effect of VR on patient satisfaction during and after regional anesthesia. As you described in the limitation section, this study is not a RCT but an observational study. Among the patients the 50 who first completed the questionnaire were included in the analysis. This suggest serious selection bias. Patient who have a high satisfaction have a possibility of completing the questionnaire. In terms of design and ethical issues, this can be done as RCT.

This is a noninvasive study and could have been done under RCT.
However, this was done under before and after study and the patient selection was conducted with the first come first served for questionnaires
This may rise patient selection bias.

Reviewer 2 Report

Dear Editor,

this is an extremely well written paper.

However, I just have one concern:

Page 4 ethical issue: Is it correct, that this study on patients falls into the categoreies of "Research studies not involving human subjects"?

Reviewer 3 Report

Compliments to the authors of this study for an innovative approach to making an anxiety provoking experience less stressful and possibly more enjoyable through the use of technology.

A few questions/suggestions:

In 1. Introduction: bottom of page one: "Virtual reality (VR) appears to be one of these effective diversion methods, having proved to be worth especially in..."  Should this word be worthwhile? Something different? In 2. Methods: It is implied that VR was not previously available before Jan 14th; however, later, under statistical analysis, there is reference to four patients who had previously received VR.  The authors need to clarify is VR was actually unavailable, or specify how this was defined.  3 patients received intraoperative sedation- how much? Were they still given the postoperative survey immediately upon arrival to the PACU? This needs to be clarified.  Section 2.7 Statistical analysis- again, refers to preliminary satisfaction data from four patients managed with a VR device....again, if this was part of the study design, I am a bit unclear about the language used to define "Available" in previous descriptors of the study. Table 1: States 100% of VR patients received an axillary block, as opposed to a distal block. Strange to have that many patients receive the same type of block (and probably deserving of comment in reference to the 2 vs 1 patient who required supplemental sedation).  Ulnar nerve block is misspelled.  Section 3.2  Again, in discussion regarding postoperative satisfaction score being higher in VR group- this deserves comment regarding the 2 vs 1 patients who received intraoperative sedation and how their satisfaction scores compared to their non-sedated counterparts. 4. Discussion: Why is the word anxiolytic placed within <<>>? 4 Discussion, pgh 3: in the sentences: "They have shown a non-significant sedation-sparking effect in the virtual reality (could use VR here) group but no benefit in terms of postoperative patient satisfaction.  However, in this small study, each of the two groups had systematically a sedation protocol (missing a verb???- "received?")..." 4 Discussion. pgh 4: 1st sentence is grammatically awkward.

Should be noted that Figures 2 and 3 will print non discriminatory on in greyscale, and perhaps some other colors could be considered. Figure 2, Month 2 is missing a box - intentional?

Overall, I think this paper is worthwhile and may encourage users to consider incorporating VR technology into the OR environment - agree further large scale studies should be pursued.